# Foot-and-Mouth Disease Virus Serotype O Exhibits Phenomenal Genetic Lineage Diversity in India during 2018–2022

**DOI:** 10.3390/v15071529

**Published:** 2023-07-10

**Authors:** Shyam Singh Dahiya, Saravanan Subramaniam, Jajati Keshari Mohapatra, Manoranjan Rout, Jitendra Kumar Biswal, Priyabrata Giri, Vinayak Nayak, Rabindra Prasad Singh

**Affiliations:** ICAR-National Institute on Foot and Mouth Disease, International Centre for FMD, Arugul, Bhubaneswar 752050, India; shyam.dahiya@icar.gov.in (S.S.D.); jajati.mohapatra@icar.gov.in (J.K.M.); manoranjan.rout@icar.gov.in (M.R.); jitendra.biswal@icar.gov.in (J.K.B.); priyabratagiri5566@gmail.com (P.G.); shradhakarnayak1234@gmail.com (V.N.); rabindra.singh@icar.gov.in (R.P.S.)

**Keywords:** FMDV, India, serotype O, genetic lineages, antigenic relationship

## Abstract

In India, widespread foot-and-mouth disease (FMD) outbreaks occurred in 2021. The objective of this study was to identify genetic lineages and evaluate the antigenic relationships of FMD virus (FMDV) isolates gathered from outbreaks reported between 2019 and 2022. Our study shows that the lineages O/ME–SA/Ind2001e and the O/ME–SA/Cluster-2018 were both responsible for the FMD outbreaks on an epidemic scale during 2021. This observation is in contrast to earlier findings that suggested epidemic-scale FMD outbreaks in India are often connected to a single genetic lineage. Additionally, we report here the identification of the O/ME–SA/PanAsia-2/ANT10 sub-lineage in India for the first time, which was connected to two intermittent outbreaks in Jammu and Kashmir. The current study demonstrates that the O/ME–SA/ind2001e lineage has a strong presence outside of the Indian subcontinent. Furthermore, the O/ME–SA/Cluster-2018 was observed to have a wider geographic distribution than previously, and like the O/ME–SA/Ind2001d and O/ME–SA/Ind2001e lineages in the past, it may eventually spread outside of its geographic niche. For O/ME–SA/Ind2001e and O/ME–SA/Cluster-2018, the predicted substitution rate for the VP1 region was 6.737 × 10^−3^ and 8.257 × 10^−3^ nt substitutions per site per year, respectively. The time of the most recent common ancestor of the O/ME–SA/Ind2001e and O/ME–SA/Cluster-2018 strains suggests that the viruses possibly emerged during 2003–2011 and 2009–2017, respectively. Recent sightings of the O/ME–SA/PanAsia2/ANT10 virus in India and the O/ME–SA/Ind2001e virus in Pakistan point to possible cross-border transit of the viruses. The results of a two-dimensional viral neutralization test revealed that all of the field isolates were antigenically matched to the currently used Indian vaccine strain O INDR2/1975. These results suggest that the serotype O vaccine strain can protect against outbreaks brought on by all three circulating lineages.

## 1. Introduction

Foot-and-mouth Disease (FMD) is one of the main challenges to the livestock sector in India and other endemic nations. The causative agent, FMD virus (FMDV), has seven immunologically different serotypes: O, A, C, Asia 1, and Southern African Territories (SAT) 1, SAT 2, and SAT 3. These serotypes can be further divided into several topotypes, lineages, sub-lineages, and an ever-expanding ensemble of variants [1]. Three of the seven serotypes, including serotypes O, A, and Asia 1, are prevalent in India. Multiple serotypes, associated antigenic diversity, short-lived vaccine-induced immunity, rapid transmission, and other associated factors make the disease control difficult. In India, FMD outbreaks are primarily caused by serotype O (which accounts for approximately 90% of FMD outbreaks), which also dominates in other parts of the world [2]. Serotypes A and Asia1 on the other hand are associated with sporadic occurrence of FMD. The serotype O isolates thus far collected from India belong to the Middle East–South Asia (ME–SA) topotype and were further subdivided into a number of genetic lineages [3].

An official vaccination-based FMD control program has been in place in the country since 2003. The overall disease incidence and clinical sickness have progressively declined. However, there have been waves of epidemics sweeping the country cyclically once every 3–5 years, generally linked to a new genetic cluster of serotype O. For instance, O/ME–SA/PanAsia, O/ME–SA/Ind2001d, and O/ME–SA/Ind2001e, respectively, were responsible for the FMD epidemics in 2007, 2013, and 2018 [4,5]. Further, the lineages O/ME–SA/Ind2001d and O/ME–SA/Ind2001e were reported to cause FMD outbreaks in other countries in the Middle East, East Asia, South East Asia, North Africa, and the Indian sub-continent [6].

FMDV exhibits a high level of genetic diversity and generates a population of variants with related sequences as a result of accumulated mutations and/or recombination [7]. The FMDV genome (approximately 8.4 kb in length) contains a 5′ untranslated region (UTR), followed by a single open reading frame, and a short flanking 3′ UTR. The polyprotein is processed by viral proteases, resulting in four structural (VP1, VP2, VP3, and VP4) and ten non-structural (L, 2A, 2B, 2C, 3A, 3B1, 3B2, 3B3, 3C, and 3D) proteins [8]. Among the structural proteins, hyper-variable VP1 is considered to be an important genotype determinant due to the presence of two loop structures, G-H and B-C, on the viral surface. Therefore, the VP1 coding sequence has been extensively employed in studies on the evolutionary dynamics of FMDV, which is necessary for comprehending the epidemiological patterns of these viruses and identifying potential sources of outbreaks [9]. Earlier, we reported the genetic and antigenic characterization of FMDV isolates collected during 2014–2018 [5], which revealed the emergence of novel genetic lineages including O/ME–SA/Ind2001e and O/ME–SA/Cluster-2018. In 2021, several FMD outbreaks were reported in India, three years after the last epidemic wave in 2018. In this study, we aim to characterize the FMDV isolates sampled during 2019–2022, by VP1 sequence-based evolutionary analyses, and vaccine matching, with due emphasis on the strains associated with the FMD epidemic in 2021.

## 2. Materials and Method

### 2.1. Virus Isolation

During 2019–2022, the state FMD network laboratories collected a total of 3756 clinical samples from suspected FMD cases across various states and union territories (UTs) in India. First, the samples were tested for serotype identification using in-house sandwich ELISA, and ELISA-negative samples were further tested using reverse transcription multiplex PCR. Subsequently, the samples were inoculated into a BHK-21 cell monolayer for virus isolation. Infected cell cultures were harvested after complete cytopathic effects were observed. The infected cell culture supernatants were used for vaccine matching and sequence-based studies. In addition, sequences were also generated directly from clinical materials for samples that could not be isolated in cell culture.

### 2.2. VP1 Sequencing

Following the manufacturer’s instructions, the RNeasy Mini Kit (Qiagen, Hilden, Germany) was used to extract viral RNA from 138 infected cell culture supernatants and clinical samples. Reverse transcription was carried out using MMLV reverse transcriptase (Promega, Madison, WI, USA) and NK61 primer [10]. All the PCR amplification was performed using pfu DNA polymerase (Fermentas, Waltham, MA, USA). For the amplification of VP1, the primer combination of ARS4 [10] and NK61 were used. The details of PCR, the primer used for sequencing, and the thermal profile were similar to those previously mentioned [11]. The PCR products were purified using a QIAquick PCR Purification Kit (Qiagen, Germany), and the amplicons were sequenced using an ABI 3130 Genetic Analyzer (Applied Biosystems, Waltham, MA, USA).

### 2.3. Phylogenetic Analysis

In addition to the VP1 sequences of 138 strains generated in this study, we retrieved 223 Indian VP1 sequences from the GenBank and the Institute Genetic Database. The sequences were aligned using the MUSCLE tool. The mean and pairwise divergence were then computed. To further explore the differences between the genetic lineages and the Indian vaccine strain O/INDR2/1975, nucleotide and deduced amino acid sequences were compared using the BioEdit program version 7.2.5.0 [12]. To assess the evolutionary relationships among FMDV isolates, phylogenetic trees were inferred by the maximum likelihood (ML) method based on the nucleotide alignment of the VP1 sequences using the MEGA software v. 11 [13]. The ML phylogeny was produced under the Kimura 2-parameter evolution model with rate variation following a gamma distribution as determined by the model finder, and the robustness of the tree topology was assessed by bootstrap analysis with 1000 iterations.

### 2.4. Phylodynamic Analysis

In order to explore the evolutionary characteristics of recent FMDV isolates, 542 full-length VP1 sequences from India and other countries were used for phylodynamic analyses. The presence of a temporal signal was examined by root-to-tip regression with Tempest v.1.5.3 software [14]. The Markov chain Monte Carlo (MCMC) method was performed in BEAST v.1.10.4 [15], and a relaxed and uncorrelated lognormal clock and exponential coalescent population prior were used to estimate the temporal phylogeny and substitution rate. Three independent runs of 200 million generations were carried out, their convergence was evaluated, and the log and tree files were then combined with the aid of Log Combiner. A maximum clade credibility (MCC) tree was summarized using Tree Annotator v1.10.4, with the burn-in option used to remove the first 10% of sampled trees, and the resulting tree was visualized by FigTree v 1.4.4. Phylogeographic analyses were performed, using an asymmetric substitution model with BSSVS options to infer asymmetric diffusion rates [16] between any pairwise location state and allowing BF calculations to verify significant diffusion rates.

### 2.5. Selection Pressure Analysis

Three likelihood approaches were employed to determine the positive selection pressure at certain codon sites: the single likelihood ancestor counting (SLAC) method, the fixed effects likelihood (FEL) method, and a Bayesian strategy called FUBAR. The ratio of non-synonymous (dN) to synonymous (dS) substitutions per site (ratio: dN/dS) was used to calculate the strength of selection pressure. In general, posterior probabilities > 0.9 for FUBAR and *p* < 0.1 for SLAC strongly imply positive selection. The Mixed Effects Model of Evolution (MEME) was used to identify the codon sites that were the subject of episodic diversifying selection. At significance levels (*p* 0.05), strong evidence of selection was accepted. All the analyses were carried out using the online Datamonkey web server [17].

### 2.6. Vaccine Matching Analysis

Monovalent bovine vaccinal serum (BVS) against Indian serotype O vaccine strain O/INDR2/1975 was obtained from the serum repository of ICAR-NIFMD, Bhubaneswar, India. Before testing, the serum was inactivated at 56 °C for 30 min in a water bath. Vaccine matching was performed using a two-dimensional virus neutralization test as described by [18]. The antibody titer was determined as the reciprocal of the last dilution of serum that neutralized 100 TCID50 in 50% of the wells. The relationship value (r1-value) was calculated as a ratio of antibody titers against field isolates to those against the vaccine strain, averaged from the two separate runs. An adequate antigenic homology between a field isolate and the vaccination strain is indicated by an r1-value of ≥0.3. On the other hand, the r1-value of less than 0.3 indicates an antigenic divergence.

## 3. Results

### 3.1. Serotype Detection and Virus Isolation

During 2019–2022, a total of 532 FMD outbreaks were serotype confirmed, of which 489 were caused by serotype O, accounting for about 92% of the total FMD occurrences in the country (Table 1). Compared to 2019 and 2020, approximately a six-fold increase in the number of outbreaks was observed in 2021. The outbreaks were extensively reported from several states and UTs during 2021. Apparently, the surge in the number of outbreaks in 2021 was due to serotype O, which was responsible for 92% of the total FMD outbreaks reported. Moreover, serotype O was found to be responsible for 98%, 83%, and 93% of the FMD outbreaks, respectively, in 2019, 2020, and 2022. Serotypes A and Asia1 were associated with sporadic incidences. In total, 3756 clinical samples were processed for serotype identification using sandwich ELISA and RT-mPCR, which revealed serotype O in the majority of the FMD-positive samples (*n* = 1502). The clinical materials were passaged in BHK-21 cells, and FMDV could be isolated from 190 samples, of which 165 were confirmed to be serotype O.

### 3.2. Phylogenetic Relationships

Earlier, we reported the genetic characterization of FMDV serotype O isolates collected during 2014–2018 from India (5). In the current study, 13 isolates collected during 2018 have also been sequenced, in addition to those sequenced during 2019–2022. In total, 138 isolates sequenced in this study (Appendix A) and 223 Indian sequences retrieved from the public domain and institute data bank were used for comparative analyses. Phylogenetic analysis based on the full-length VP1 region showed that all FMDV strains collected in India between 2018 and 2022 could be divided into three distinct lineages, O/ME–SA/Ind2001e, O/ME–SA/Cluster-2018, and O/ME–SA/PanAsia-2, with supporting bootstrap values of 99%, 84%, and 99%, respectively (Figure 1 and Appendix A). Out of 138 strains sequenced in this study, 78 clustered within the O/ME–SA/Ind2001e lineage, whereas 56 isolates grouped within the O/ME–SA/Cluster-2018. Interestingly, four isolates grouped together within the O/ME–SA/PanAsia-2, sharing descent with ANT10 sub-lineage, which has never been identified in India so far.

These PanAsia-2 strains were isolated from two outbreaks reported from Jammu and Kashmir in 2021, shared ancestry, and demonstrated 96% sequence homology with an isolate collected from Pakistan in 2019 (Figure 2). Unfortunately, sequences collected after 2019 from the region are not available in the public domain for inclusion in the comparison. The inclusion of a larger number of sequences of recent origin from this group will shed more light on the possible transmission pattern. The O/ME–SA/PanAsia-2/ANT10 showed pairwise mean genetic distances of 10.1 and 12.4% from O/ME–SA/Cluster-2018 and O/ME–SA/Ind2001e, respectively, at the nucleotide level. The O/ME–SA/Cluster-2018 and the O/ME–SA/Ind2001e differed by 12.6% in the mean distance. The O/ME–SA/Cluster-2018 shared ancestry with the O/ME–SA/PanAsia-2. With the currently used Indian serotype O vaccine strain INDR2/1975, the isolates of these three genetic groups showed a mean genetic distance of 12.1 to 13.5%.

FMDV isolates (*n* = 138) sequenced in this study were collected from 79 outbreaks (on many occasions, more than one virus isolate from the same outbreak was sequenced). FMD strains associated with 64 outbreaks during 2021 were characterized phylogenetically, which revealed the involvement of the O/ME–SA/Ind2001e lineage in 36 outbreaks, the O/ME–SA/cluster-2018 in 26 outbreaks, and two outbreaks were caused by O/ME–SA/PanAsia-2 (Figure 3). Though it was not possible to sequence all the FMDV outbreak strains due to the non-receipt of samples or inappropriate quality of samples leading to failure in PCR amplification, it can be presumed that the FMD epidemic observed in 2021 was due to both the O/ME–SA/Ind2001e lineage (56 percent of the outbreaks) and the O/ME–SA/cluster-2018 (41 percent of the outbreaks). On some occasions, from a single outbreak, both O/ME–SA/Ind2001e and O/ME–SA/Cluster 2018 lineage could be detected. For instance, an outbreak in the states of Karnataka (January 2021), Maharashtra (August 2021), and Jammu and Kashmir (July 2021) was caused by the simultaneous involvement of both lineages.

### 3.3. Phylodynamic Analyses

In this analysis, 542 serotype O VP1 sequences, comprising 361 Indian and 181 foreign sequences obtained from GenBank (accessed on 24 January 2023), were included. India is predicted to be the ancestral root state for O/ME–SA/Ind2001e and O/ME–SA/Cluster-2018 in the current analysis with good statistical support (root state posterior probabilities = 0.99). For the O/ME–SA/PanAsia-2/ANT10 sub-lineage, Pakistan had the highest root-state posterior probabilities of 0.99 (Figure 4). The O/ME–SA/ind2001e lineage has been reported from Bangladesh, Bhutan, Nepal, Myanmar, Thailand, the United Arab Emirates, Saudi Arabia, Russia, Mauritius, China, South Korea, and Jordan during 2015–2018 [6,19]. The O/ME–SA/Ind2001e has firmly established itself outside of the Indian subcontinent, as this study further demonstrates. Recently, outbreaks of O/ME–SA/Ind2001e lineage were reported in Pakistan in 2019 [20]. The outbreaks in Pakistan were due to three phylogenetically distinct groups closely related to strains circulating in Nepal, India, and Bhutan (pp = 1). Out of three clades of Ind2001e detected in Pakistan, two clades share a close relationship with Indian isolates (MRCA December 2017, PP = 1, and February 2018, PP = 0.99), and one with isolates from Bhutan (MRCA October 2017, PP = 0.01).

The O/ME–SA/Cluster-2018 was found to be associated with FMD outbreaks in the state of Maharashtra during 2019 and 2020. Subsequently, the lineage was identified in the states of Karnataka, Odisha, Jammu and Kashmir, Jharkhand, Bihar, and Sikkim in 2021 and 2022. Outside India, the lineage was first detected in Bangladesh in 2021. The O/ME–SA/Cluster-2018 isolates found in Bangladesh shared a close genetic relationship with their Indian counterparts (MRCA August 2020, PP-0.91). Apart from this, none of the isolates from other countries compared in this study were grouped within O/ME–SA/cluster-2018. The scenario may change if a larger number of sequences are made publicly available from the neighboring countries. Though they represent clades distinct from each other with a pp of 0.99, the O/ME–SA/cluster-2018 and O/ME–SA/PanAsia-2 share a common ancestor with MRCA dating back to 2010.49.

Within India, frequent exchanges of virus strains between the states were observed (Figure 5). The state of Karnataka and the UT of Jammu and Kashmir might have played an important role in seeding virus dissemination for O/ME–SA/Ind2001e and O/ME–SA/cluster-2018, respectively. The unregulated animal movement between the states plays an important role in the wide dissemination of FMDV outbreaks in the country.

### 3.4. Lineage O/ME–SA/Ind2001e

The dataset contains 432 sequences, 288 of which were collected in India and the rest from other countries. The isolates of the O/ME–SA/Ind2001e lineage showed a maximum and mean genetic divergence of 4.7 and 3.0% at the nucleotide level and 3.8 and 2.0% at the amino acid level, respectively. Overall, in the nucleotide alignment, 339 sites were found to be invariable and 300 sites showed polymorphisms. Out of 300 polymorphic nucleotide sites, 220 were parsimony informative and 80 were singleton variable sites. The substitution rate for the VP1 region was estimated to be 6.737 × 10^−3^ (95% HPD range 5.539–8.014 × 10^−3^) substitution/site/year, with a predicted time to a most recent common ancestor (tMRCA) of 2007 (95% HPD: 2003–2011). The relative nucleotide substitution rates at all three codon positions in VP1 showed that substitutions were more frequent at the third codon position (2.024, 95% HPD 1.885–2.164) compared to the first (0.536, 95% HPD 0.453–0.697) and second (0.399, 95% HPD 0.306–0.499), as expected.

The dN/dS ratio for the Ind2001e isolates was estimated to be 0.190, signifying evidence of negative selection in shaping their evolution. Further, negative Tajima’s *D* values (−2.07566) and low nucleotide diversity (0.02869) among the VP1 coding region Ind2001e lineage viruses point to a population expansion after a recent selective sweep or bottleneck. To support this, Fu and Li’s D* test statistic (−4.15352) and F* test statistic (−3.57142) also showed negative values. The Tajima’s *D* values, and Fu and Li’s D* test statistic are associated with statistical significance. Further evidence for negative selection is identified at 75 codon positions. Only three codon positions (96, 172, and 176) were found to be under pervasive positive selection by SLAC (*p* < 0.1), and selection pressure at site 96 was also identified by FUBAR (pp > 90%). The MEME likelihood approach was used to identify sites under episodic selection, and nine codon positions (14, 15, 43, 76, 96, 172, 176, 190, and 209) were found to be under episodic selection pressure (*p* < 0.5). The approach projected episodic pressure at codon positions 112 and 201.

### 3.5. Lineage O/ME–SA/Cluster-2018

The dataset comprised VP1 sequences from 73 isolates, of which 69 isolates were sampled in India and only four isolates were collected in Bangladesh in 2021. The pairwise nucleotide and amino acid divergence among the cluster 2018 was determined to be 2.3–6.8% and 1.4–3.7%, respectively, with a mean divergence of 3 and 1%. Overall, in the nucleotide alignment, 510 sites were found to be invariable and 129 sites showed polymorphisms. Out of 129 nucleotide sites, 83 were parsimony informative and 46 were singleton variable sites. In total, 99 codon positions out of 213 were found conserved. The dN/dS ratio for O/ME–SA/Cluster-2018 was estimated to be 0.127, indicating strong purifying selection and evidence for purifying selection was found at nine codon positions. Further, negative Tajima’s *D* values (−1.15314) and low nucleotide diversity (0.02821) among the VP1 coding region of O/ME–SA/Cluster-2018 viruses point to population expansion after a recent selective sweep or bottleneck. To support this, Fu and Li’s D* test statistic (−1.64861) and F* test statistic (−1.73722) also showed negative values. The Tajima’s *D* values and Fu and Li’s D* test statistics are not statistically significant. No evidence of diversifying selection was found in the data set by SLAC, FEL, and FUBAR. Many codon sites may experience selection in a restricted number of branches, designated as episodic diversifying selection. The MEME likelihood approach was used to identify sites under episodic selection. The approach projected episodic pressure at codon positions 112 and 201. The MRCA of O/ME–SA/Cluster-2018 isolates was dated to 2014 (95% HPD 2009–2017). The rate of evolutionary change for the VP1 coding region was estimated to be 8.257 × 10^−3^ nt substitutions per site per year (95% HPD, 3.766 × 10^−3^ to 1.26 × 10^−2^ nt/site-year). The relative nucleotide substitution rates at all three codon positions in VP1 showed that substitutions were more frequent at the third codon position (2.245, 95% HPD 2.017–2.467) compared to the first (0.556, 95% HPD 0.347–0.761) and second (0.20, 95% HPD 0.093–0.317), as expected.

### 3.6. Variations in Amino Acid

Comparison of the deduced amino acid sequences of O/ME–SA/Cluster-2018 isolates to vaccine strain O INDR2/1975 revealed variations at 11 positions (P4T, A96K, L126M, D138E, G139S, S140H, V141A, N143S, I144V, A158T, and N197S) at consensus sequence level (Figure 6). With respect to the sub-lineage O/ME–SA/Ind2001e, the amino acid substitutions at eight positions (P4T, A13T, D138E, S140A, I144V, A158T, N197E, and E198Q) as reported earlier are maintained (5). The PanAsia-2/ANT10 appears to be relatively more divergent from vaccine strain O INDR2/1975, as they showed variations at 14 positions (P4T, T101S, N133D, D138E, G139N, S140R, V141A, I144V, A158T, A915Q, N197S, A199T, V206T, and V209E). The amino acid positions (VP1-43, 44, 144, 148, 149, 154, 208) that are critical for antigenic sites were found fully conserved in all the Ind2001 isolates except for I → V replacement at position 144. All the serotype O Indian isolates sequenced so far had the same replacement (I → V), irrespective of the lineage [4]. High levels of conservation at immunologically critical sites might be due to functional and structural constraints, which partially explains the good antigenic match between the vaccine strain and field isolates. The three genetic lineages are characterized by specific signature amino acid substitutions. The amino acid changes that distinguish O/ME–SA/Ind2001e isolates from the other two lineages were found at four positions (13T, 140A, 197E, and 198Q), and for O/ME–SA/cluster-2018, exclusive changes were found at five positions (96K, 126M, 139S, 140H, and 143S). The PanAsia-2/ANT10 showed a maximum of eight specific substitutions (101S, 133D, 139N, 140R, 195Q, 199T, 206I, and 209E) compared to other lineages.

### 3.7. Vaccine Matching

In total, 59 serotype O field isolates were subjected to a vaccine-matching exercise. Field isolates were selected based on their geographical location, collection date, genetic group, and adaptation to the BHK-21 cell culture. The isolates selected represent the O/ME–SA/Ind2001e lineage (*n* = 30), O/ME–SA/Clustre-2018 (*n* = 27), and O/ME–SA/PanAsia-2/ANT10 (*n* = 2). All the isolates, irrespective of the genetic groups to which they belong, showed antigenic match (r-value > 0.3) with the currently used vaccine strain INDR2/1975 (Table 2). Both the isolates of O/ME–SA/PanAsia-2/ANT10 also showed antigenic homology with the vaccine strain.

## 4. Discussion

The prevalence of FMD in India is a major hurdle to the growth of the livestock industry due to its adverse impact on productivity, and trade in livestock and livestock products. In India, a uniform vaccine strain policy and standard vaccination strategy have been implemented countrywide. Under this program, all cattle and buffaloes are vaccinated bi-annually with an inactivated trivalent FMD vaccine for protection against FMD. Complex epidemiology of the disease poses a serious challenge to its control in FMD-endemic countries. In India, FMD is primarily caused by FMDV serotype O. FMD outbreaks on an epizootic scale were recorded in India in 2021. In this study, genetic and antigenic characterization of FMDV serotype O viruses isolated from India during 2018–2022 is reported. The VP1 coding sequence of 138 serotype O isolates and antigenic relationships of 59 isolates with the Indian vaccine strain were determined. By combining reference sequences from India and other parts of the world, the molecular epidemiology of FMDV serotype O was studied using Bayesian phylogeographic inference and the maximum likelihood method.

Our analyses revealed that all serotype O viruses collected in India belonged to the ME–SA topotype. The majority of the isolates were grouped within the O/ME–SA/Ind2001e lineage and the O/ME–SA/cluster-2018, representing 56% and 41% outbreaks, respectively. The lineage O/ME–SA/Ind2001e has circulated extensively in South East Asia, the Middle East, and East Asia since its first detection in Nepal in 2012 and has caused a series of epidemics in several countries outside of Pool 2 [6]. In India, the O/ME–SA/Ind2001e lineage was first detected in 2015, in spite of intensive surveillance [5]. The O/ME–SA/Ind2001e lineage co-circulated with the then dominant O/ME–SA/Ind2001d lineage during 2015–2017. Subsequently, an increase in the incidence of O/ME–SA/Ind2001e was observed, with the eventual disappearance of the O/ME–SA/Ind2001d lineage from the field. Our estimate of the time to the most recent common ancestor (TMRCA) of the global O/ME–SA/Ind2001e in 2007 (mean TMRCA January 2007; 95% highest posterior density [HPD], August 2002–July 2010) is comparable to prior estimates of a TMRCA during 2007 (mean TMRCA April 2007; 95% highest posterior density [HPD], March 2006–April 2008) [6].

The O/ME–SA/Cluster-2018 was first detected in 2018 in India in a limited number of outbreaks, and subsequently increase in circulation has been noticed in 2021 and 2022. After its first detection in the state of Maharashtra, O/ME–SA/Cluster-2018 lineage has spread to six more Indian states. Almost 40% of the FMD outbreaks observed in 2021 were caused by this lineage. Outside India, the lineage was detected in Bangladesh in 2021, which indicates that the lineage is progressively undergoing geographic expansion. The Bayesian phylogeographic analysis provided strong support for India as the country of origin for O/ME–SA/Ind2001e and O/ME–SA/cluster-2018. Nonetheless, it should be highlighted that sample biases that affect phylogeographic reconstruction quality, especially in areas with sparse data, may have an impact on the precise origins of epidemics [6].

The recent documentation of O/ME–SA/PanAsia-2/ANT10 in India and O/ME–SA/Ind2001e in Pakistan clearly indicates virus exchange between the two countries. Though transmission and incursion of FMDV were reported in countries such as Bangladesh and Nepal, the exchange of FMDV between India and Pakistan is buzzing as the two countries do not indulge in direct livestock trade [20], and also have strict border control. Since the lineage was detected in Jammu and Kashmir, a UT bordering Pakistan, the possibility of airborne transmission cannot be excluded. The inclusion of recent sequences of PanAsia-2 will give more insights into the evolutionary and transmission events. FMDV-O topotype ME–SA and lineage PanAsia-2 are the extensively distributed lineages in Iraq, Iran, Afghanistan, and Pakistan, and the dominant sub-lineage of serotype O that is commonly detected in Pakistan is O/ME–SA/PanAsia-2/ANT10 [21]. Animal movements and international trade are considered important risk factors for the emergence of various exotic FMDVs, including serotypes and lineages that are not included in vaccination plans [22]. On many occasions, virus sequences from different regions and states of India clustered together closely, indicating frequent inter-state transmission of FMDV. This chain of transmission events is mainly caused by the unregulated movement of infected animals, as reported in previous studies [4,5]. This altogether highlights the huge risk potential of the transboundary nature of FMDV and its ability to easily spread across the country.

In India, higher FMD incidences on the epidemic scale occur cyclically after every few (3–5) years, and generally, a new genetic lineage of serotype O is associated with such an upsurge. Surprisingly, the FMD epidemic observed in 2021 was caused by two lineages, including O/ME–SA/Ind2001e and O/ME–SA/Cluster-2018. Another interesting observation is that the previous three epidemics occurred in a gap of 5–6 years, but the 2021 epidemic happened within three years after the last epidemic in 2018. This could be due to the fact that during 2019 and 2020, vaccinations were not practiced due to COVID-19-linked lockdown and movement restrictions. The FMD epidemics in Sri Lanka generally appear every four to six years, although the evolutionary dynamics of the patterns of appearance remain unclear [23].

FMDV evolves primarily by point mutation, and the size of this mutant swarm has a significant impact on the virus’s ability to adapt, spread, and cause disease [24]. The appearance of new virus strains can pose a challenge to disease control strategies in endemic settings, especially when the in-use vaccine strain fails to offer sufficient cross-protection [25]. In the vaccine matching analysis, all the serotype O isolates showed an r-value of >0.3, which indicates a perfect antigenic match with the field isolates. The antigenic relationship between the field strain and the vaccine strain established through vaccine matching studies is a critical determinant of the efficacy of the vaccine strain used. However, other important factors such as regularity of vaccination schedule, vaccine coverage, antigenic mass in the vaccine, cold chain logistics from the factory to field, and biosecurity breaches at the outbreak site, including unrestricted animal movement, might contribute to the number and extensiveness of outbreaks in the country. Generally, serotype O vaccines are broadly cross-reactive, often exhibiting in vitro protection against a number of genotypes. For instance, in South America, the O/Campos vaccine has been used for routine vaccination for more than five decades and still matches the viruses circulating in the region [26]. The Indian type O vaccine strain (O/IND/R2/1975) has also been reported to have antigenic similarities with viruses from East Africa, mainly Eritrea, Ethiopia, Kenya, and Sudan [27]. The strain O/INDR2/1975 has been in use for more than four decades in India and is still able to provide cross-protection to different genetic variants within the ME–SA topotype circulating in the country.

## 5. Conclusions

The increased number of outbreaks of FMD in 2021 was associated with two lineages, viz., O/ME–SA/Ind2001e and O/ME–SA/Cluster-2018. Proportionately based on the sequences analyzed, it can be presumed in simplistic terms that 60% of the FMD outbreaks might have been caused by the O/ME–SA/Ind2001e lineage, with 40% caused by the O/ME–SA/Cluster-2018. Two isolated FMD outbreaks in Jammu and Kashmir was caused by O/ME–SA/PanAsia-2/ANT10, and seems to be self-limiting. The PanAsia-2 strain could not be detected further in any of the FMD outbreaks recorded subsequently. The O/ME–SA/Cluster-2018 appears to be the next dominant lineage, as previously anticipated [5], and it may eventually spread outside of its geographic niche of origin, much like the O/ME–SA/Ind2001d and e lineages in the past. Irrespective of genetic diversity, vaccine matching analyses established clear antigenic homology between the field isolates and the currently used Indian vaccine strain.

## Figures and Tables

**Figure 1 viruses-15-01529-f001:**
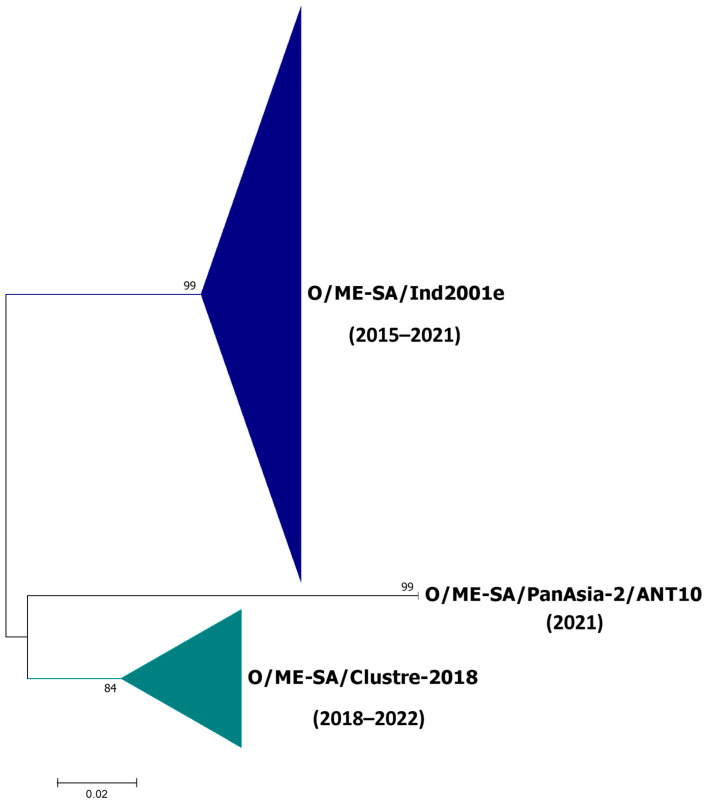
The phylogenetic relationships of the 361 FMDV serotype O isolates, including 138 isolates sequenced in this study and collected from India between 2015 and 2022, are shown. Bootstrap values for 1000 replicates are indicated. The expanded maximum likelihood tree is provided in Appendix A.

**Figure 2 viruses-15-01529-f002:**
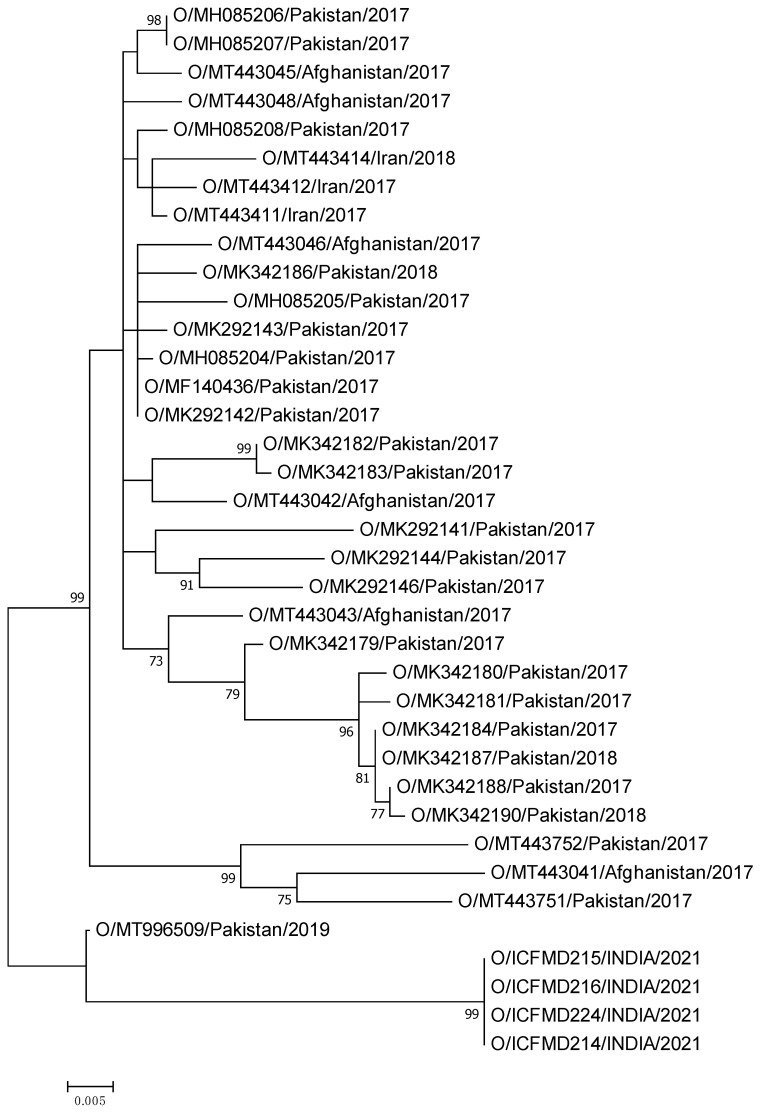
Phylogenetic relationships of the O/ME–SA/PanAsia-2/ANT10 lineage, reported for the first time in India, and inferred using the maximum likelihood method are shown. Four isolates collected from Jammu and Kashmir shared ancestry with an isolate sampled in 2019 from Pakistan.

**Figure 3 viruses-15-01529-f003:**
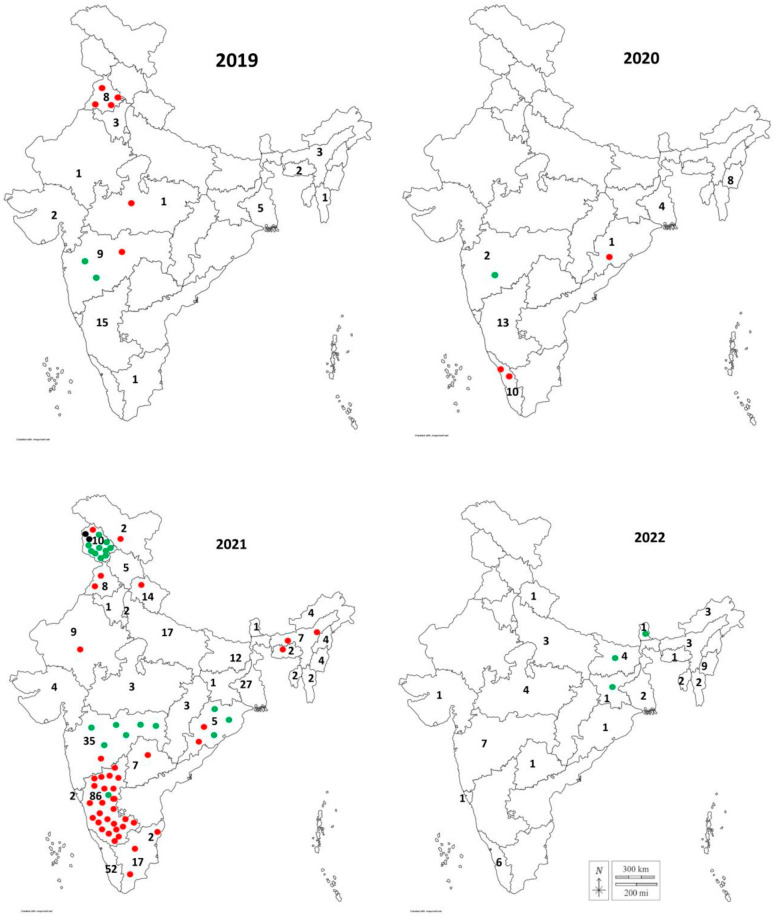
The spatial distribution of different lineages of serotype O in India during 2019–2022. Red dots denote the O/ME–SA/Ind2001e lineage; green dots indicate the O/ME–SA/Cluster-2018; and blue dots specify the O/ME–SA/PanAsia-2/ANT10 sub-lineage. The trend clearly indicates an increase in circulation for O/ME–SA/Cluster-2018. The numbers indicate the number of FMD outbreaks caused by serotype O recorded in each state.

**Figure 4 viruses-15-01529-f004:**
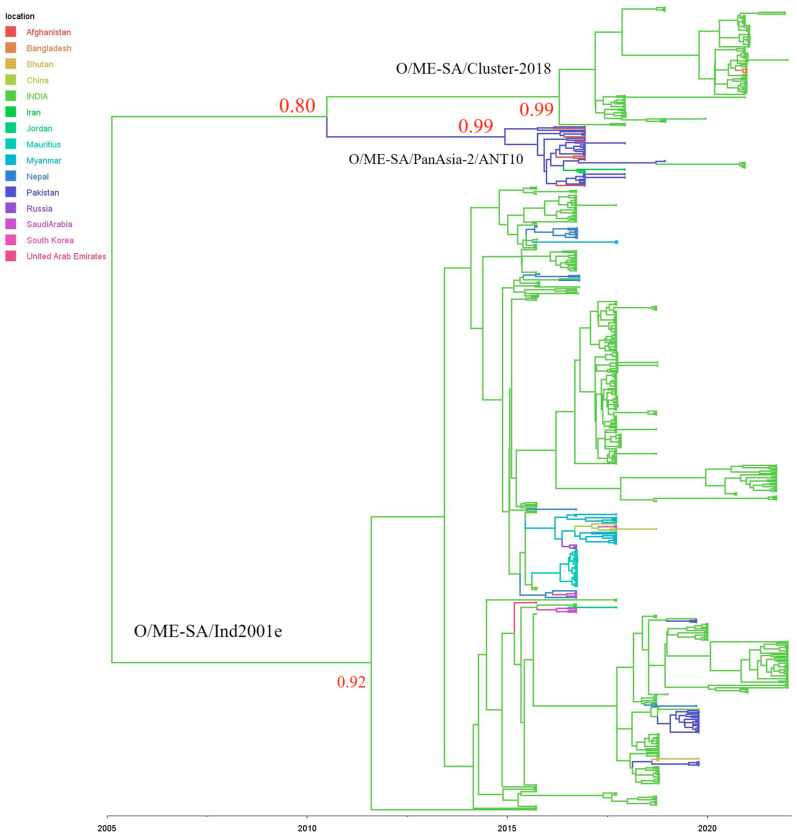
Bayesian time-scaled phylogeny of FMDV serotype O with inferred geographical location states. Maximum clade credibility tree of FMDV serotype O viruses based on complete VP1 coding sequences inferred using BEAST. Branch lengths are scaled according to time, as indicated by the horizontal axis. Branch colors denote inferred location states, as shown in the color key.

**Figure 5 viruses-15-01529-f005:**
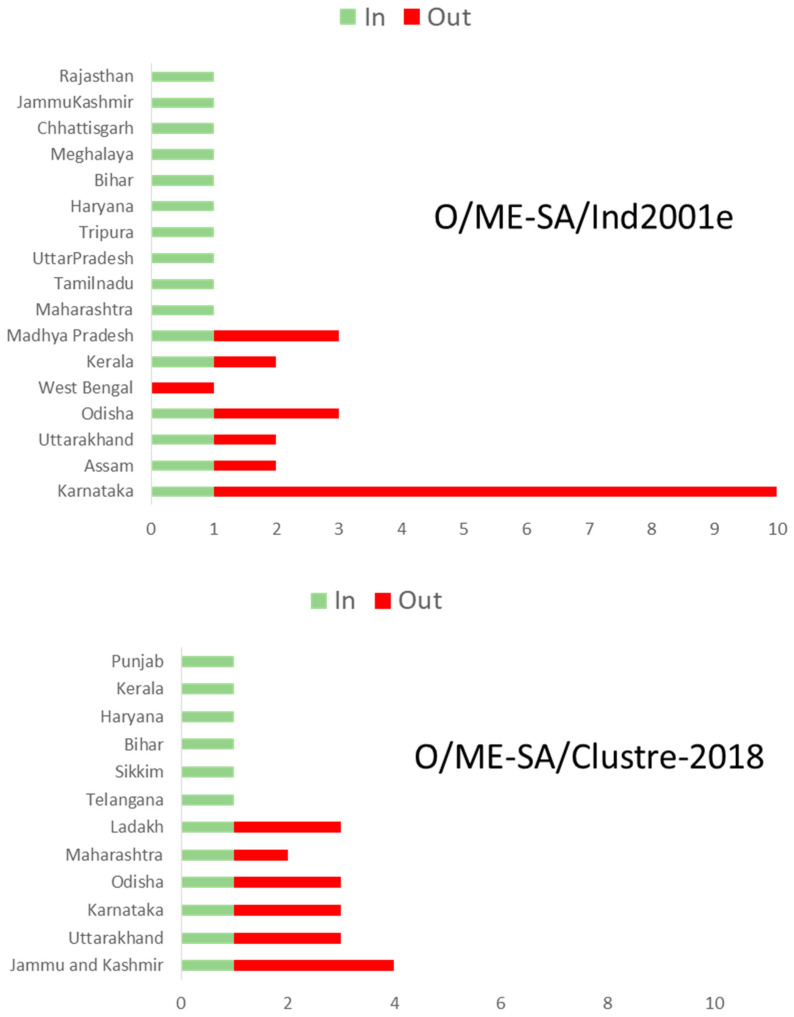
Histogram of the total number of location-state transitions inferred from two major FMD serotype O lineages currently circulating in India. Only the strongly supported transitions (BF > 10 and posterior probability > 0.4) are shown. This indicates frequent virus exchange among the states.

**Figure 6 viruses-15-01529-f006:**
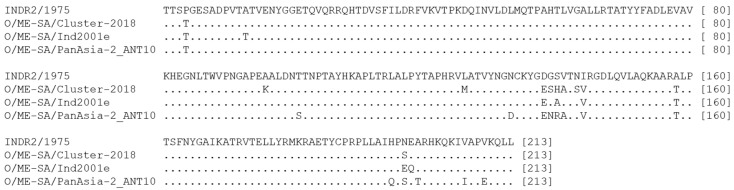
Predicted amino acid sequences of three different FMDV lineages currently prevailing in India display consensus changes in the VP1 region compared to the currently used serotype O vaccine strain INDR2/1975.

**Table 1 viruses-15-01529-t001:** Number of FMD outbreaks caused by serotype O during 2019–2022 in India.

Year	Total Number of Clinical Samples Tested	Number of Samples Positive for Serotype O	Total Number of FMD Outbreaks	Number of Outbreaks by Serotype O
2019	306	145	52	51
2020	215	94	46	38
2021	2824	1122	378	349
2022	411	141	55	51

**Table 2 viruses-15-01529-t002:** One-way antigenic relationship value of FMDV Serotype O isolates.

S No	Isolate ID	r-Value	Genetic Group
1	O/PD38/2018/Uttarakhand	0.47	O/ME–SA/Cluster-2018
2	O/PD57/2018/Haryana	0.33	O/ME–SA/Cluster-2018
3	O/PD224/2018/Kerala	0.42	O/ME–SA/Cluster-2018
4	O/ICFMD202/2018/Tamilnadu	0.71	O/ME–SA/Cluster-2018
5	O/ICFMD218/2018/Tamilnadu	0.59	O/ME–SA/Cluster-2018
6	O/ICFMD/184/2021/Jammu and Kashmir	0.93	O/ME–SA/Cluster-2018
7	O/ICFMD/189/2021/Jammu and Kashmir	1	O/ME–SA/Cluster-2018
8	O/ICFMD/244/2021/Jammu and Kashmir	1	O/ME–SA/Cluster-2018
9	O/ICFMD/345/2021/Jammu and Kashmir	0.467	O/ME–SA/Cluster-2018
10	O/ICFMD/357/2021/Jammu and Kashmir	1	O/ME–SA/Cluster-2018
11	O/ICFMD/441/2021/Jammu and Kashmir	1	O/ME–SA/Cluster-2018
12	O/ICFMD/271/2021/Ladakh	1	O/ME–SA/Cluster-2018
13	O/ICFMD/280/2021/Ladakh	1	O/ME–SA/Cluster-2018
14	O/ICFMD/285/2021/Ladakh	1	O/ME–SA/Cluster-2018
15	O/ICFMD/291/2021/Ladakh	1	O/ME–SA/Cluster-2018
16	O/ICFMD/302/2021/Ladakh	0.606	O/ME–SA/Cluster-2018
17	O/ICFMD/344/2021/Punjab	1	O/ME–SA/Cluster-2018
18	O/ICFMD/366/2021/Punjab	1	O/ME–SA/Cluster-2018
19	O/ICFMD/369/2021/Punjab	1	O/ME–SA/Cluster-2018
20	O/ICFMD/350/2021/Maharashtra	0.5	O/ME–SA/Cluster-2018
21	O/ICFMD/513/2021/Maharashtra	1	O/ME–SA/Cluster-2018
22	O/ICFMD/522/2021/Maharashtra	1	O/ME–SA/Cluster-2018
23	O/ICFMD/958/2021/Maharashtra	0.834	O/ME–SA/Cluster-2018
24	O/ICFMD/850/2021/Rajasthan	1	O/ME–SA/Cluster-2018
25	ICFMD/47/2022/Jharkhand	1	O/ME–SA/Cluster-2018
26	ICFMD/160/2022/Sikkim	0.901	O/ME–SA/Cluster-2018
27	ICFMD/182/2022/Bihar	1	O/ME–SA/Cluster-2018
28	O/PD214/2018/HimachalPradesh	0.31	O/ME–SA/Ind2001e
29	O/PD400/2018/Uttarakhand	0.53	O/ME–SA/Ind2001e
30	O/PD403/2018/Uttarakhand	0.51	O/ME–SA/Ind2001e
31	O/ICFMD19/2019/Punjab	0.71	O/ME–SA/Ind2001e
32	O/ICFMD21/2019/Punjab	0.43	O/ME–SA/Ind2001e
33	O/ICFMD22/2019/Punjab	0.52	O/ME–SA/Ind2001e
34	O/ICFMD/241/2021/Odisha	1	O/ME–SA/Ind2001e
35	O/ICFMD/531/2021/Maharashtra	0.655	O/ME–SA/Ind2001e
36	O/ICFMD/472/2021/Tamilnadu	1	O/ME–SA/Ind2001e
37	O/ICFMD/534/2021/Tamilnadu	0.841	O/ME–SA/Ind2001e
38	O/ICFMD/548/2021/Karnataka	1	O/ME–SA/Ind2001e
39	O/ICFMD/556/2021/Karnataka	0.501	O/ME–SA/Ind2001e
40	O/ICFMD/572/2021/Karnataka	1	O/ME–SA/Ind2001e
41	O/ICFMD/591/2021/Karnataka	0.88	O/ME–SA/Ind2001e
42	O/ICFMD/594/2021/Karnataka	1	O/ME–SA/Ind2001e
43	O/ICFMD/606/2021/Karnataka	0.726	O/ME–SA/Ind2001e
44	O/ICFMD/622/2021/Karnataka	0.72	O/ME–SA/Ind2001e
45	O/ICFMD/643/2021/Karnataka	0.785	O/ME–SA/Ind2001e
46	O/ICFMD/658/2021/Karnataka	0.9	O/ME–SA/Ind2001e
47	O/ICFMD/669/2021/Karnataka	0.649	O/ME–SA/Ind2001e
48	O/ICFMD/688/2021/Karnataka	0.88	O/ME–SA/Ind2001e
49	O/ICFMD/689/2021/Karnataka	1	O/ME–SA/Ind2001e
50	O/ICFMD/755/2021/Karnataka	0.841	O/ME–SA/Ind2001e
51	O/ICFMD/770/2021/Karnataka	0.72	O/ME–SA/Ind2001e
52	O/ICFMD/803/2021/Karnataka	1	O/ME–SA/Ind2001e
53	O/ICFMD/820/2021/Karnataka	0.9	O/ME–SA/Ind2001e
54	O/ICFMD/828/2021/Karnataka	1	O/ME–SA/Ind2001e
55	O/ICFMD/829/2021/Karnataka	0.856	O/ME–SA/Ind2001e
56	O/ICFMD/830/2021/Karnataka	0.85	O/ME–SA/Ind2001e
57	OICFMD/890/2021/Assam	0.494	O/ME–SA/Ind2001e
58	O/ICFMD/214/2021/Jammu and Kashmir	0.878	O/ME–SA/PanAsia-2/ANT10
59	O/ICFMD/224/2021/Jammu and Kashmir	0.834	O/ME–SA/PanAsia-2/ANT10

## Data Availability

All required data are available as texts and figures in the main text of the article or in the Appendix A. The sequence data sets generated during this research are publicly available at NCBI GenBank.

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
