# Peer review of "Foot-and-Mouth Disease Virus Serotype O Exhibits Phenomenal Genetic Lineage Diversity in India during 2018–2022"

_viruses, 2023, doi:10.3390/v15071529_

Round 1
Reviewer 1 Report
The manuscript written by Shyam Singh Dahiya et al about FMD is a very good one and gives information to readers in the area. Over all I found it nice and publishable in its current format. As a suggestion for the authors I have two points to mention
The first one is : In doing the phylodynamics couldn’t be good to see sequences from others areas in addition to the Asian countries sequences. Of course your interest is only O/MANISA of other regions and other O strains.
My second comment and suggestion is on the vaccine matching test. Here the author mentioned that the vaccine can protect against outbreaks brought on by all three circulating lineages. Is that possible to do prediction of protection with your r-value without challenge of the natural host.
My second comment and suggestion is on the vaccine matching test. Here the author mentioned that the vaccine can protect against outbreaks brought on by all three circulating lineages. Is that possible to do prediction of protection with your r-value with out challenge of the natural host.
